# Low-Intensity Exercise Training Additionally Increases Mitochondrial Dynamics Caused by High-Fat Diet (HFD) but Has No Additional Effect on Mitochondrial Biogenesis in Fast-Twitch Muscle by HFD

**DOI:** 10.3390/ijerph17155461

**Published:** 2020-07-29

**Authors:** Yun Seok Kang, Donghun Seong, Jae Cheol Kim, Sang Hyun Kim

**Affiliations:** Department of Sports Science, College of Natural Science, Jeonbuk National University, 567 Baekje-daero, Deokjin-gu, Jeonju-si 54896, Korea; kangys53@jbnu.ac.kr (Y.S.K.); hun3030@jbnu.ac.kr (D.S.); Kjc@jbnu.ac.kr (J.C.K.)

**Keywords:** high-fat diet, exercise, mitochondrial function, mitochondrial dynamics, mitochondrial biogenesis

## Abstract

This study examines how the high-fat diet (HFD) affects mitochondrial dynamics and biogenesis, and also whether combining it with low-intensity endurance exercise adds to these effects. Six 8-week-old male Sprague–Dawley (SD) rats were put on control (CON; standard chow diet), HF (HFD intake), and HFEx (HFD + low-intensity treadmill exercise) for 6 weeks. As a result, no change in body weight was observed among the groups. However, epididymal fat mass increased significantly in the two groups that had been given HFD. Blood free fatty acid (FFA) also increased significantly in the HF group. While HFD increased insulin resistance (IR), this was improved significantly in the HFEx group. HFD also significantly increased mitochondrial biogenesis-related factors (PPARδ, PGC-1α, and mtTFA) and mitochondrial electron transport chain proteins; however, no additional effect from exercise was observed. Mitochondrial dynamic-related factors were also affected: Mfn2 increased significantly in the HFEx group, while Drp1 and Fis-1 increased significantly in both the HF and HFEx groups. The number of mitochondria in the subsarcolemmal region, and their size in the subsarcolemmal and intermyofibrillar regions, also increased significantly in the HFEx group. Taken overall, these results show that HFD in combination with low-intensity endurance exercise has no additive effect on mitochondrial biogenesis, although it does have such an effect on mitochondrial dynamics by improving IR.

## 1. Introduction

Endurance exercises lead to enhanced respiratory ability in skeletal muscles due to increases in mitochondrial enzymes [1,2] such as peroxisome proliferator-activated receptor (PPAR) γ-coactivator 1α (PGC-1α), which plays an integral role in muscular respiration [3,4]. PGC-1α transcription is regulated by various factors [5,6,7,8], including myocyte enhancer factor 2, cAMP response element, and reactive oxygen species that reside in its promoter area. According to recent studies, it is also regulated by PPARδ [9,10].

PPARδ is a ligand-activated nuclear receptor that activates the transcription of fatty acid oxidation enzymes [11,12], and is increased by free fatty acids (FFAs) [13]. In the case of specific PPARδ activation or PPARδ overexpression within skeletal muscles using transgenic mice, muscular mitochondria increase without the post-transcriptional mechanism or an increase in PGC-1α mRNA expression [12,14]. However, PGC-1α mRNA can also be increased in skeletal muscles by short bursts of intensive exercise or by low-intensity long-term endurance exercise [15,16,17,18].

Hence, an additional effect was expected to result from combining high-fat diet (HFD) with exercise, along with their respective individual effects, due to post-transcriptional and transcriptional gene regulatory mechanisms. This hypothesis is supported by a study of Fillmore et al. [19] that has shown additive effects between HFD and chronic activation of AMP-activated protein kinase (AMPK) on mitochondrial biogenesis, compared to HFD alone. In other words, since exercise activates AMPK in conjunction with HFD it affects mitochondrial biogenesis in skeletal muscles via different mechanisms. However, Fillmore et al. [19] have activated AMPK using aminoimidazole-4-carboxamide ribonucleoside (AICAR), which mimics the effects of physical exercise by increasing glucose uptake in vivo [20] and suppressing *Acetyl*-*CoA* carboxylase (*ACC*) and fatty acid oxidation in vitro [21,22].

AICAR is not only listed as a prohibited substance by the World Anti-Doping Agency (http://www.wada-ama.org/), but it also cannot act as a substitute for the effects of actual exercise due to its side effects, such as its promotion of protein degradation via increased expression of E3 ligases [23,24].

Since exercise and nutrient stimuli regulate specific mitochondrial dynamics (biogenesis, fission, fusion, and mitophagy) [25,26], it is important to examine how physical exercise, rather than drugs that mimic their effects, in combination with HFD affects mitochondrial-specific dynamics in skeletal muscles in terms of health or performance enhancement. This is because mitochondria perform a variety of functions, including energy homeostasis [26], reactive oxygen species signaling, apoptosis-programmed cell death [27], immune signaling [28], and cellular metabolism regulation [29], while mitochondrial dysfunction is related to many diseases, such as type 2 diabetes and muscular dystrophy [26]. Additionally, in cases where performing effective exercises to enhance mitochondrial function is difficult, due to muscular atrophy from aging or musculoskeletal damage, a parallel HFD treatment regime can serve as an effective alternative.

Accordingly, this study examines whether the combination of HFD and exercise affects specific mitochondrial dynamics, and whether these effects are caused by different mechanisms, by investigating the effects of HFD in combination with exercise in comparison to the effects of HFD only.

## 2. Materials and Methods

The 8-week-old (approximately 250 g) male Sprague–Dawley (SD) rats (Damul Science Inc., Daejeon, South Korea) were used in this study. All of the rats were given a week-long period to acclimatize to their new environment, during which they had unlimited access to food (carbohydrate, 58.9%; fat, 12.4%; protein, 28.7%; Purina corp., St. Louis, MO, USA) and water. The colony was maintained at a temperature of 21 °C and a humidity level of 40–60%. Light and dark periods alternated every 12 h.

Following acclimatization, rats were randomly assigned to three groups: control (CON; *n* = 6), normal diet and sedentary; HF (*n* = 6), 45% HFD intake; HFD + low-intensity treadmill exercise (HFEx; *n* = 6), 45% HFD intake and low-intensity endurance treadmill exercise. HFD chow was purchased from Research Diets Inc. (carbohydrate, 35%; fat, 45%; protein, 20%; D12451, New Brunswick, NJ, USA). Low-intensity endurance exercise was performed on a motor treadmill at a speed of 5–10 m/min for 30–45 min for 3 times/week for the first week and 15 m/min for 60 min/day for the subsequent five weeks. After 6 weeks of the HFD and exercise regime, rats were anesthetized using sodium pentobarbital (50 mg/kg of body weight; Fort Dodge Animal Health, Overland Park, KS, USA) and tissues were collected. The extensor digitorum longus (EDL) and soleus muscles were extracted at 18 h after the final training session. The left-side muscle was rapid frozen and stored at −80 °C until analysis by Western blotting, while the right-side muscle was fixed for mitochondrial size and number analysis using transmission electron microscopy (TEM). Epididymal fat pads were excised and the weight was measured to determine body fat mass. The present study received approval from the Institutional Animal Care and Use Committee of Jeonbuk National University (IACUC approval no. CBNU-2017-0008).

### 2.1. Blood Lipids

Blood samples were separated by centrifugation at 3000 rpm for 15 min at 4 °C. Plasma levels of free fat acid (FFA), glucose, and insulin were measured using ELISA kit (MyBioSource Inc., San Diego, CA, USA).

Based on fasting insulin and glucose levels the homeostasis model assessment as an index of insulin resistance (HOMA-IR) was calculated using the following formula [30]:HOMA-IR = Fasting insulin [μU/mL] × Fasting glucose [mg/dL]/405

### 2.2. Western Blotting

The frozen tissues were ground into powder with homogenization after adding RIPA buffer containing protease and phosphatase inhibitors. Protein concentrations were determined using the BCA method with bovine serum albumin as a standard. Of the protein samples 20 μg were separated by 8–12% SDS-PAGE and then transferred to nitrocellulose membranes. The membranes were incubated overnight at 4 °C in 5% skim milk with the following primary antibodies: β-actin (Sigma, St. Louis, MO, USA); NADH ubiquinone oxidoreductase (NADH-UO), Cytochrome C (Cyto C) (Invitrogen, Waltham, MA, USA); PPARδ, mitochondrial transcription factor A (mtTFA), mitofusin 2 (Mfn2), dynamin-related protein-1 (Drp1), mitochondrial fission 1 (Fis1; Santa Cruz Biotechnology, Dallas, TX, USA); cytochrome c oxidase subunit-Ι (COX I), COX IV, and PGC-1α (ABCam, Cambridge, UK). After extensive rinsing with PBST, the blots were incubated with secondary goat anti-rabbit or goat anti-mouse antibody at room temperature for 1 h. Band visualization was performed using an ECL Western Blotting Detection Reagent (GE Healthcare, RPN2232, Chalfont St Giles, UK) and ChemiDoc XRS + (BIO RAD, Hercules, CA, USA).

### 2.3. TEM

For TEM, the extracted EDL muscle was immediately fixed in a solution containing 2.5% glutaraldehyde and 4% formaldehyde in 0.1 M phosphate buffer [31] at pH 7.4 for 2-h. The muscle were post-fixed for two hours in 1% osmium tetroxide, dehydrated in a graded series of ethanol, and embedded in Epon-812 resin mixed by Luft’s method [32]. Semithin sections, cut in both transverse and longitudinal planes, were stained with 0.1% toluidine blue for light microscopy. Thin sections (around 80 nm thick) for transmission electron microscopy were cut in both transverse and longitudinal planes with a NOVA ultramicrotome (LKB, Vienna, Austria), and picked up on 100-mesh grids. After staining with uranyl acetate and lead citrate, the specimens were viewed with an electron microscope (H7650, accelerating voltage, 80 kV, Hitachi, Japan). Mitochondrial number and size were measured using image analysis computer software (Analysis pro ver. 3.2, Soft Imaging System GmbH, Hamburg, Germany).

### 2.4. Statistical Analysis

Statistical analysis was performed using Sigma Stat software (SigmaPlot 12.0, San Jose, CA, USA). Data are presented as mean ± standard error of the mean (SEM). The change in body weight was analyzed by two-way ANOVA with repeated measures. One-way ANOVA comparison of fat mass, blood parameters, expression patterns of proteins related to mitochondrial biogenesis, and function. A Bonferroni’s (two-way) or LSD (one-way) post hoc test was conducted to determine the significance when appropriate. All data were checked for normality and equal variances between groups. Statistical significance level was set at *p* < 0.05.

## 3. Results

### 3.1. Body Weight and Epididymal Fat Mass

Figure 1 shows changes in body weight and epididymal fat mass over a 6-week treatment. While the body weights of all the rats increased over the course of the experiment, there was no difference among groups (Figure 1A). HFD and HFEx groups increased post-treatment epididymal fat mass per body weight significantly compared to CON group (all *p* < 0.05; 0.054 ± 0.004 vs. 0.034 ± 0.002; 0.047 ± 0.004 vs. 0.034 ± 0.002, respectively), with no significant differences between HFD and HFEx groups (Figure 1B).

### 3.2. Blood Parameters

Table 1 shows changes in blood lipid components due to the 12-h food deprivation period after the final exercise session during a 6-week treatment. FFA increased significantly (*p* < 0.05) in HF compared to CON. While it decreased somewhat in the HFEx group compared to HF, there was no significant difference. Insulin increased in HF compared to CON, but there were no significant differences across the groups overall. Glucose and HOMA-IR increased significantly (*p* < 0.05) in HF but decreased significantly (*p* < 0.05) in the HFEx group compared to the HF group.

### 3.3. Mitochondrial Biogenesis-Related Protein Content

Figure 2 shows changes in expression levels of protein involved in mitochondrial biogenesis over 6-week treatment. In the HF and HFEx groups, PPARδ, PGC-1α, and mtTFA increased significantly compared to the CON group (*p* < 0.05), with no significant differences between HFD and HFEx groups.

### 3.4. Mitochondrial Electron Transport Chain (ETC) Protein Content

Figure 3 shows changes in expression patterns of proteins involved in mitochondrial electron transport chain over a 6-week treatment. In the HF and HFEx groups, NADH-UO, Cyto C, COX-I, and COX-IV increased significantly compared to the CON group (*p* < 0.05), with no significant differences between HFD and HFEx groups.

### 3.5. Mitochondrial Dynamic-Related Protein Content

Figure 4 shows changes in expression patterns of protein related to mitochondrial fusion and fission over the 6-week treatment. Mfn2, a protein that induces mitochondrial fusion, increased significantly (*p* < 0.05) only in the HFEx group compared to the CON group. Drp1 and Fis-1 increased significantly (*p* < 0.05) in the HF and HFEx groups compared to the CON group, with no significant differences between HFD and HFEx groups.

### 3.6. Mitochondria Number and Size

Figure 5 shows whether HFD and low-intensity endurance exercises cause differences in the mitochondrial number and size patterns over and above differences in protein expression patterns related to mitochondrial dynamics. Figure 5A shows a recording of mitochondria within skeletal muscles using TEM, on which the mitochondrial number and size are based, as shown in Figure 5B. The mitochondrial number increased significantly (*p* < 0.05) in the subsarcolemmal region of the HFEx group compared to that of the CON group. The mitochondrial size increased significantly (*p* < 0.05) in the subsarcolemmal and intermyofibrillar regions of the HFEx group compared to the CON and HF groups.

## 4. Discussion

We investigated whether HFD increases the number and function of mitochondria in EDL muscles via PGC-1α and PPARδ and, if the mitochondrial biosynthesis mechanisms caused by exercise and HFD are not the same, whether combining a HFD with exercise induces additive effects on the mitochondria in skeletal muscles, compared to the individual effects of HFD. As a result, 6-week HFD increased IR, this was improved in the HFEx group. Additionally, in the HF and HFEx groups, expression of genes related to mitochondrial biogenesis and mitochondrial dynamic was increased, with no significant differences between HFD and HFEx groups. However, the mitochondrial number and size by TEM were only increased in the HFEx group.

A long-term HFD induces insulin resistance (IR). Since people with IR have been found to have 30% reduced skeletal muscle mitochondria [33], this reduction due to HFD [34] has been posited as the cause of IR [35,36,37]. However, numerous studies have shown that reduced mitochondria is not in itself a direct cause of IR [13,38,39,40]; rather, HFD or a high plasma fatty acid concentration increases expression of mitochondrial enzymes related to the fatty acid oxidation pathway, the citrate cycle, and the respiratory chain in skeletal muscles and PPARδ and PGC-1α [13,19,40]. Contrary to the common belief that HFD is always undesirable and that its effects are exclusively negative, it can be used as a means of increasing mitochondrial activity within skeletal muscles.

Since mitochondria play a central role in cell life and death, they are related to fatality rates in certain diseases [41]. For this reason, maintaining and managing both the number and functioning of mitochondria are highly important in healthcare. Increasing the number and functioning of mitochondria can be achieved via expression and activation regulation of the transcription and co-transcription factors related to mitochondrial dynamics or biogenesis. These processes are regulated by numerous factors, including exercise, nutrition, and cold [42,43,44]. In particular, certain forms of endurance exercise, such as running or swimming, and HFD increase PGC-1α expression [40,45], a co-transcription factor that plays an integral role in mitochondrial biosynthesis, as well as PPARδ, a transcription factor [13,46]. However, the mitochondrial biosynthesis mechanisms resulting from combining exercise with HFD do not clearly overlap with one another [47].

First, our study found that 6-week of an HFD regime did not increase weight, but visceral fat increased significantly in 8-week-old rats. However, while the amount of increased visceral fat due to HFD did not differ greatly between the HF and HFEx groups, there was a tendency for the degree of increase to decrease once exercise was added. Hence, it is postulated that negative changes in body composition due to an HFD can be avoided by calibrating the intensity, duration, and amount of exercise in relation to the quantity and duration of the diet. Mitochondrial function is related to IR [48,49,50,51]. In order to test whether different treatments lead to IR, HOMA-IR was computed based on blood glucose and insulin concentrations. While it increased significantly in the HF group, it decreased significantly in the HFEx group. These results show that the low-intensity treadmill exercise performed in this study improved HFD-induced IR; this is in line with a previous study on humans [52] that showed that low-intensity and low-volume exercises also improve IR. However, IR was calculated using the formula from Matthews et al. [30] in this study. Therefore, it is desirable to present the result by insulin tolerance test and/or intraperitoneal glucose tolerance test, not by the formula, in order to reach a more conclusive conclusion.

PPARδ, which is related to PGC-1α and known to play an integral role in mitochondrial biogenesis, is increased by FFA [13,47]. The HFD used in this study also increased FFA concentration. In order to examine whether this increase in FFA also increases PPARδ, protein expression was measured in skeletal muscles, and was found to have increased by 2.5 times. HFD is also known either to decrease or have no effect on the mRNA [19,53] or protein expression of PGC-1α [54]. However, in this study, however, HFD increased PGC-1α protein, a result that has been replicated in several other studies [19,40,55]. It is thought that PPARδ increased due to a post-transcriptional mechanism or PGC-1α protein degradation, as described in Koh et al. [47]. PGC-1α activates nuclear respiratory factor (NRF), which in turn increases expression of mitochondrial transcription factor A (mtTFA), thereby increasing expression of mitochondrial enzymes such as ATP synthetase and cytochrome c oxidase (COX) subunits (COX-I and COX-IV) [56].

Based on the findings of this study, it is thought that the increased expression of mtTFA and mitochondrial enzymes led to increased expression of mitochondrial enzymes in HFD mice via the PPARδ–PGC-1α–mtTFA axis in the fast-twitch muscle EDL, but not in slow-twitch muscle soleus (data not shown). A similar increase was observed in the HFEx group, which means that adding exercise to the HFD regime does not produce any additive effect on mitochondrial biogenesis via the PPARδ–PGC-1α–mtTFA axis, in comparison with HFD treatment alone. In other words, exercise had no effect on mitochondrial biogenesis. Since the present study posited a case in which exercise to improve mitochondrial function cannot be effectively performed, because of muscular atrophy or musculoskeletal damage due to aging, a treadmill program was designed with a very low intensity and volume (5–10 m/min for 30–45 min for the first week and 15 m/min for 60 min/day for the subsequent five weeks, three times/week). It remains a possibility that the intensity and/or duration of the exercise regime were insufficient to meet the minimum requirements for mitochondrial biogenesis enhancement. Therefore, further research is required to set the minimum exercise frequency, intensity, and time for mitochondrial biogenesis to occur.

Mitochondria are highly dynamic organelles in which constant movement, fusion, and fission occur [57,58]. Their morphology is maintained by the balance between mitochondrial fusion and fission [59].

While mitofusins (Mfn1 and Mfn2) and optic atrophy-1 (OPA1) are key actors in fusion, dynamin-related protein-1 (Drp1), mitochondrial fission 1 (Fis-1) induces fission [60,61,62]. HFD regulates the expression of these factors. According to Liu et al. [63], a four-week HFD decreases Mfn1 and 2 and increases Fis1 and Drp1 to reduce mitochondrial function. Another study found differences based on the type of fat involved: whereas a saturated fat increased Drp1 and Fis1 to reduce mitochondrial functioning, an unsaturated fat increased Mfn1 and 2 to enhance mitochondrial functioning [50,64].

By contrast, Silvestri et al. [55] have shown that while a four-week HFD decreases Mfn2, it produced no change in the activity of the mitochondrial respiratory complexes (I, II, and IV). Additionally, Leduc-Gaudet et al. [65] have shown that a short-term HFD consisting primarily of lard increases both Mfn1 and Fis1 and enhances mitochondrial FA-oxidation ability, without altering mitochondrial content. While there is a difference in the duration of the HFD regime in the study by Leduc-Gaudet et al. [65] compared to the present one, the results are similar: Mfn2 was not increased significantly by HFD, but it was increased in the HFEx group, while Drp1 was increased in both the HF and HFEx groups. Moreover, Fis1 was increased in both the HF and HFEx groups, leading to increases in fusion and fission with HFD. However, image analysis using TEM showed that the number of intermyofibrillar mitochondria increased in the HFEx group, while the sizes of both the subsarcolemmal and intermyofibrillar mitochondria increased in the HFEx group. Therefore, it is thought that combining HFD with low-intensity exercise induces more positive results in mitochondrial dynamics than results with HFD alone. According to a recent study, mitochondrial dynamics are reduced in HFD-induced IR [48,49,50,51].

However, when the duration of HFD was short enough to avoid IR, not only do Mfn2 and Fis1 increase but Drp1 content also increases, without affecting the Mfn2 and OPA1 content, thereby enhancing mitochondrial function [65]. Therefore, it is thought that HFD-induced IR was improved by treatment with HFD in combination with exercise, inducing positive effects on the mitochondrial dynamics of the HFEx group.

## 5. Conclusions

In conclusion, a six-week HFD induces IR by increasing visceral fat, rather than body weight. However, an HFD increased mitochondrial biosynthesis via the PPARβ-PGC-α-mtTFA axis. Although combining an HFD with low-intensity endurance exercise did not produce an additive effect on mitochondrial biosynthesis, it nonetheless did improve HFD-induced IR by increasing mitochondrial fusion and fission (dynamic) and enhancing mitochondrial functioning. Therefore, performing low-intensity endurance exercises in the absence of IR, which can be facilitated by adjusting the HFD treatment period, method (intermittently), or fat type (saturated or unsaturated fatty acids), can provide an alternative when an effective exercise of moderate-to-high intensity cannot be performed to ameliorate reduced mitochondrial function due to muscular atrophy or musculoskeletal damage.

## Figures and Tables

**Figure 1 ijerph-17-05461-f001:**
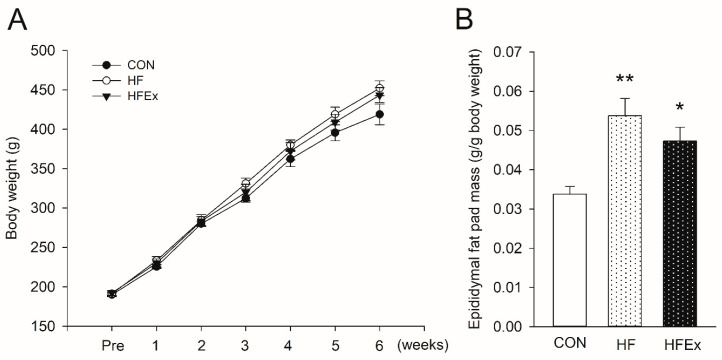
The impact of high-fat diet (HFD) or HFD and exercise. Changes in body weight (**A**) and epididymal fat pad mass (**B**) over 6-week treatment. Data are presented as means ± SEM. Data was analyzed by a two-way ANOVA with repeated measures (**A**) and one-way ANOVA (**B**). * *p* < 0.05 vs. control (CON), ** *p* < 0.001 vs. CON. CON, control; HF, HFD intake; HFEx, high-fat diet with low-intensity endurance exercise.

**Figure 2 ijerph-17-05461-f002:**
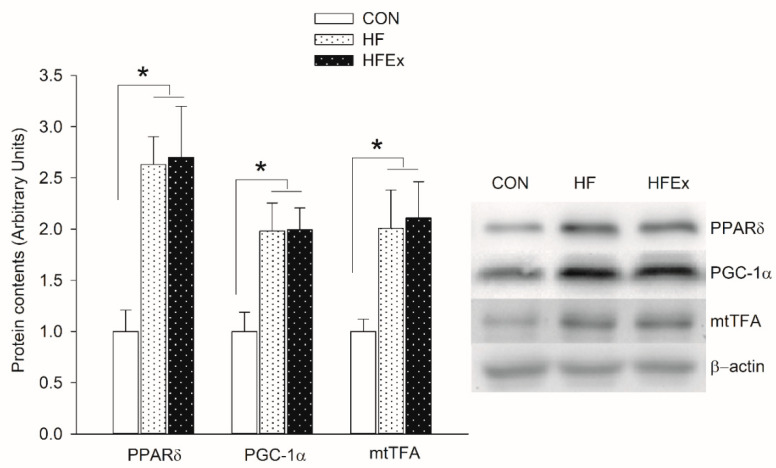
Expression levels of protein involved in mitochondrial biogenesis in the extensor digitorum longus (EDL) muscle. Data are presented as means ± SEM. Data was analyzed by one-way ANOVA. * *p* < 0.05. CON, control; HF, HFD intake; HFEx, high-fat diet with low-intensity endurance exercise; EDL, extensor digitorum longus.

**Figure 3 ijerph-17-05461-f003:**
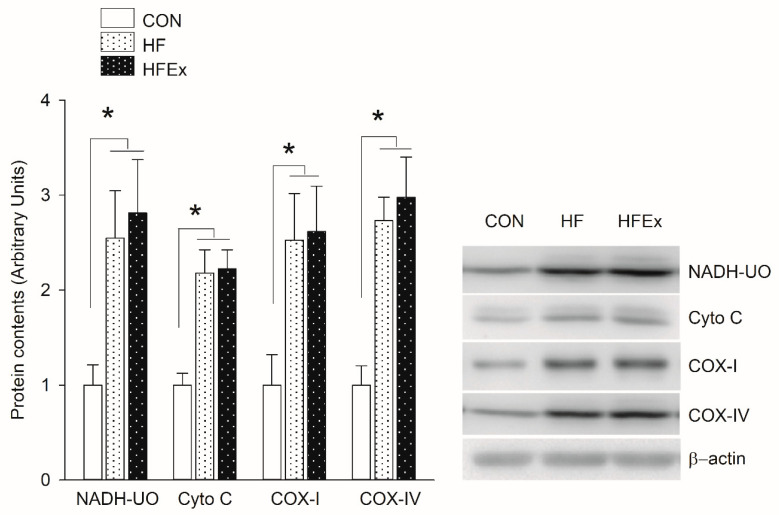
Expression levels of protein involved in mitochondrial electron transport chain in the EDL muscle. Data are presented as means ± SEM. Data was analyzed by one-way ANOVA. * *p* < 0.05 vs. CON. CON, control; HF, HFD intake; HFEx, high-fat diet with low-intensity endurance exercise; EDL, extensor digitorum longus.

**Figure 4 ijerph-17-05461-f004:**
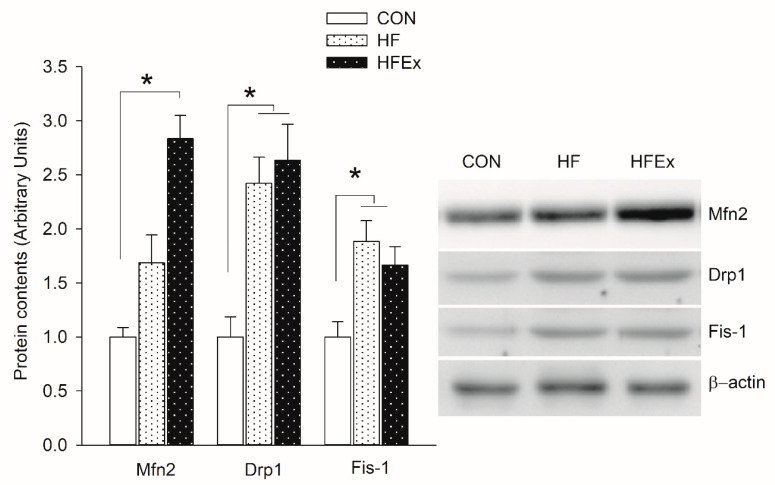
Expression levels of protein involved in mitochondrial dynamic in the EDL muscle. Data are presented as means ± SEM. Data was analyzed by one-way ANOVA. * *p* < 0.05 vs. CON. CON, control; HF, HFD intake; HFEx, high-fat diet with low-intensity endurance exercise; EDL, extensor digitorum longus.

**Figure 5 ijerph-17-05461-f005:**
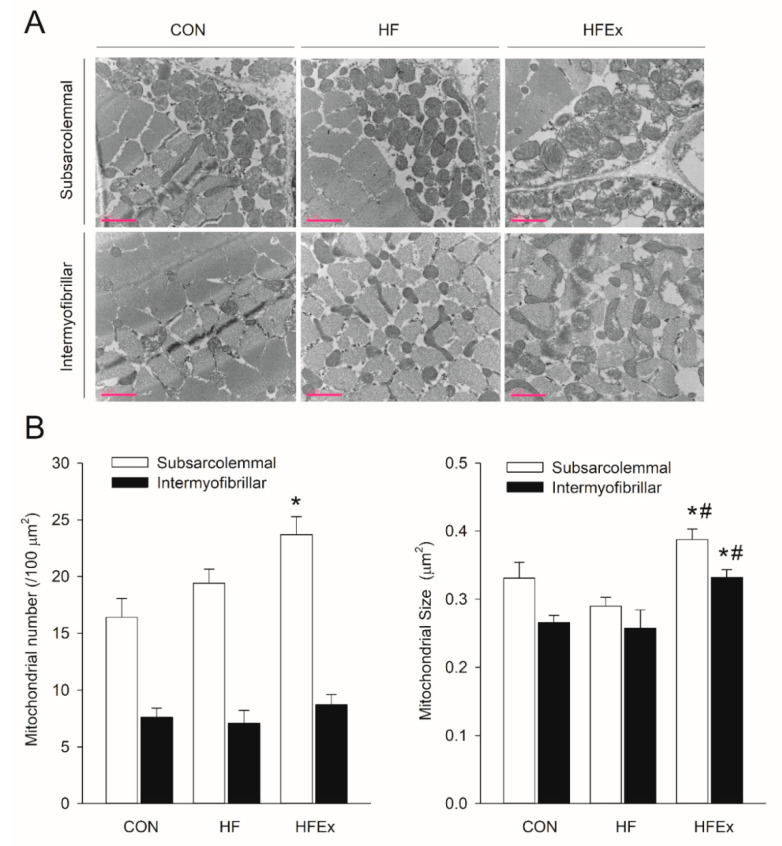
Effects of HFD or HFD and exercise on mitochondrial number and size in the EDL muscle. (**A**) Representative transmission electron micrograph (TEM) images of the rat EDL muscle. Scale bar = 2 μm. (**B**) Summary data of mitochondrial number per area and the size of individual mitochondria. Data are presented as means ± SEM. Data was analyzed by a one-way ANOVA. * *p* < 0.05 vs. CON, ^#^
*p* < 0.05 vs. HF. CON, control; HF, HFD intake; HFEx, high-fat diet with low-intensity endurance exercise; EDL, extensor digitorum longus.

**Table 1 ijerph-17-05461-t001:** Average level of blood parameters at conclusion of 6 weeks treatment.

Variables	CON	HF	HFEx
FFA (μEq/L)	508.2 ± 36.5	806.3 ± 87.3 *	618.8 ± 55.1
Insulin (μg/L)	0.30 ± 0.002	0.37 ± 0.03	0.32 ± 0.02
Glucose (mg/dL)	93.7 ± 4.5	117.1 ± 6.2 *	99.8 ± 5.9 ^#^
HOMA-IR	1.48 ± 0.11	2.36 ± 0.28 *	1.71 ± 0.24 ^#^

Data are presented as means ± SEM. Data was analyzed by one-way ANOVA. * *p* < 0.05 vs. CON, ^#^
*p* < 0.05 vs. HF. CON, control; HF, HFD intake; HFEx, high-fat diet with low-intensity endurance exercise.

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
