# Peer review of "Low-Intensity Exercise Training Additionally Increases Mitochondrial Dynamics Caused by High-Fat Diet (HFD) but Has No Additional Effect on Mitochondrial Biogenesis in Fast-Twitch Muscle by HFD"

_ijerph, 2020, doi:10.3390/ijerph17155461_

Round 1

Reviewer 1 Report

Review of “Low-intensity exercise training additionally increases mitochondrial dynamics caused by high-fat diet (HFD) but has no additional effect on mitochondrial biogenesis by HFD” (ijerph-853305)

This study investigated the combination of HFD and exercise affects specific mitochondrial dynamics. This is potentially interesting; however, several problems should be solved.

  1. EDL muscle has a large proportion of fast muscles. Since there is a known difference in mitochondrial content between fast and slow muscles, a comprehensive assessment of mitochondrial function within the muscle would require a similar assessment in the soleus muscle, for example, which has a high percentage of slow muscles.
  2. The authors did not assess dietary intake. If pair-feeding was done and dietary intake was matched to the HF group, reduction in dietary intake was expected in the HFEx group, which may have been accompanied by changes in mitochondrial function as well as a reduction in visceral fat mass.
  3. In addition, please show the detail contents of HFD and Normal diet. Ex. Proportion of carbohydrate and protein.
  4. How about the difference of the muscle mass size among the groups?
  5. To investigate the insulin resistance, ITT and/or iPGTT are desirable.
  6. How about the other mitochondrial markers, such as Mfn1 and OPA1?

Author Response

Thank you for giving me the opportunity to submit a revised draft of my manuscript titled “Low-intensity exercise training additionally increases mitochondrial dynamics caused by high-fat diet (HFD) but has no additional effect on mitochondrial biogenesis by HFD” to International Journal of Environmental Research and Public Health. I appreciate the time and effort that you and the reviewers have dedicated to providing your valuable feedback on my manuscript. I am grateful to the reviewers for their insightful comments on my paper. I have been able to incorporate changes to reflect most of the suggestions provided by the reviewers. I have highlighted the changes within the manuscript.

Here is a point-by-point response to the reviewers’ comments and concerns.

Reviewer 2 Report

Dear author,

Thank you for your considerable work. Although the manuscript discusses a high-interest topic, some issues can be addressed to improve the quality and comprehensibility:

ABSTRACT

  1. The referee suggests to add a “,” after “biogenesis” in Ln 12, and also to add “and also whether”.
  2. The authors did not explain before the singles “FFA” in Ln 16. Please, add it.

INTRODUCTION SECTION

  1. The referee suggests erasing the “,” after “low-intensity” Ln 42.
  2. The authors should change “by a study by Fillmore et al” to “by a study of Fillmore et all” Ln 45.
  3. The authors should erase the “,” after “AMPK” Ln 47.

MATERIALS AND METHODS SECTION

  1. As the author presented “8-week-old” in the Abstract section, you should maintain consistently the form used to write it. Please, change it Ln 71.
  2. Throughout the Materials and Methods section, you should present the information in the same format (number or text), i.e. “two h” Ln 112 vs. “30-45 min” Ln 81 or “15 min” Ln 92.
  3. Why are the authors presenting the SEM instead of SD?
  4. Have you studied the normality of the variables? If you did, you should describe the normality analysis in the text. If you did not, you should do a non-parametric analysis, at least to compare those results with the parametric analysis. Further, you should report all of that in the “Statistical analysis section”.

RESULTS SECTION

  1. In Figure 1, you are presenting changes in two different variables in the three groups (intra-group changes). So, you should describe that in the caption. In addition, you should include the analyses performed before the P value presentation.
  2. The caption of Figure 1 should be presented correctly. In addition, the authors used before “P <0.05” Ln 126, for that reason, you should express the "p <0.05"(i.e., Ln 148, 149, 158…) in the same way. Please, review this throughout the Results section.
  3. On Ln 134, the authors should specify “Figure 1A” or "Figure 1 shows changes in weight and epididymal fat mass...". Please, check it.
  4. The referee suggests reviewing Ln 136-137. The correctly form to express a group comparison could be: "HFD and HFEx groups increased post-treatment epididymal fat mass per weight significantly compared to CON group (all p < 0.005; 0.054 + 0.004 vs. 0.034 + 0.002; 0.047 vs. 0.034 + 0.002, respectively), with no significant differences between HFD and HFEx groups”. Please, check it and change it. Also, check the same issue in Ln 157-158 and Ln 168-169.
  5. In the Results section Ln155-157, this explanation/introduction should be included in the Method section to justify the selection of these transcription factors not in the Results section. In this specific section, you have to show your findings.
  6. The referee suggests using the same beginning format for all the paragraphs in the Results section, i.e. "Figure 1 shows changes in weight and epididymal fat mass..."Ln 134. Please, review this throughout the Results section in order to be consistent in the presentation of the results.
  7. Figure 3 shows the legends in a different place. Please, change it.
  8. You should eliminate the conclusions (Ln 169-171) from the Results section. In this section you just have to mention what you found after your analyses, writing the specific information that you have not provided in the figures or tables.

DISCUSSION SECTION

  1. In the Discussion section, you must include in the first paragraph a summary of your main results, in order to introduce the readers what you found. Actually, the first paragraph of the Discussion section should be a statement of the principal findings of your study and you may include some perspective for the clinical and public fields. The reviewer suggests rewriting this paragraph trying to focus on "what" did you find and "why" these findings could be of interest. Actually, the third paragraph of the Discussion section (Ln216-225) should be placed in the first place.
  2. In the Discussion section, the referee suggests not mentioning/referring the Figures or Tables in this section, instead of that it could be useful to add the specific values of the outcomes mentioned.
  3. In the Discussion sections (Ln 221), the referee deeply recommends you to add “in 8-week-old rats”, to clarify the sample used in this study.
  4. In the Discussion section (Ln 235), the reference should be cited after “protein expression of PGC-1α.54”.
  5. In the Discussion section (Ln 236), the referee suggests reordering the connectors, i.e. “However, in this study HFD…”.
  6. In the Discussion section (Ln 242), the referee suggests replacing “based on this study” with “based on the present study or based on the findings of this study”.
  7. In the Discussion section (Ln 248), the referee suggests replacing “due to muscular atrophy” with “because of muscular atrophy”, since you used “due to” again in the same sentence.
  8. In the Discussion section, you should include a limitation section at the end of this section. Adding here, i.e. "Ln 251-253".
  9. In the Conclusion section (Ln 282), there are two dots “.” After weight. Please, check it.
  10. In summary, although the work presented is interesting, it needs some corrections in the Results section in order to clarify and present better the information. Moreover, the discussion section should be appropriately revised and rewritten following the suggestions previously presented.

Author Response

(The authors gave the same response as above.)

Round 2

Reviewer 1 Report

The authors revised well.

However, to investigate the insulin resistance, ITT and/or iPGTT are desirable and interesting. Please mentioned this point as a limitation of this study.

Author Response

Dear DR./Mr./Ms. Reviewer

Thank you for pointing this out. I agree with this comment.

Here is a point-by-point response to the reviewers’ comments and concerns.

Reviewer: However, to investigate the insulin resistance, ITT and/or iPGTT are desirable and interesting. Please mentioned this point as a limitation of this study.

Response: Thank you for this suggestion. I have added a sentence "However, IR was calculated using the formula from Matthews et al.30 in this study. Therefore, it is desirable to present the result by insulin tolerance test and/or intraperitoneal glucose tolerance test, not by the formula, in order to reach a more conclusive conclusion." in line 238-240.

Response for English editing: I asked a native English speaker for extensive English editing. However, the Assistant Editor asked to upload the revised manuscript until July 27th. So I asked the Assistant Editor to extend the submission deadline by 30 July, but I have not received a reply yet. Therefore, I submit a revised manuscript following your suggestion without English editing.

I will resubmit the completed English editing manuscript along with the Editing Certification.

I look forward to hearing from you in due time regarding our submission and to respond to any further questions and comments you may have.

Sincerely,

Sang Hyun Kim